# Bioassay Analysis and Molecular Docking Study Revealed the Potential Medicinal Activities of Active Compounds Polygonumins B, C and D from *Polygonum minus* (*Persicaria minor*)

**DOI:** 10.3390/plants12010059

**Published:** 2022-12-22

**Authors:** Rafidah Ahmad, Andi Rifki Rosandy, Idin Sahidin, Nur Syatila Ab Ghani, Normah Mohd Noor, Syarul Nataqain Baharum

**Affiliations:** 1Metabolomics Research Laboratory, Institute of Systems Biology (INBIOSIS), Universiti Kebangsaan Malaysia, Bangi 43600, Malaysia; 2University Center of Excellence for Nutraceuticals, Biosciences and Biotechnology Research Center, Bandung Institute of Technology, Bandung 40132, Indonesia; 3Department of Pharmacy, Faculty of Pharmacy, Universitas Halu Oleo, Kendari 93232, Indonesia

**Keywords:** *Polygonum minus* 1, kesum 2, polygonumins 3, anti-acetylcholinesterase 4, anti-HIV protease 5

## Abstract

Polygonumins B, C and D, derivative compounds of polygonumins A, were isolated from the stem of *Polygonum minus*. Based on NMR results, the structure of polygonumins derivatives is comprised of four phenylpropanoid units and a sucrose unit, with a similar structure to polygonumins A. However, the structural differences between polygonumins B (1), C (2) and D (3) can be distinguished based on the existence of methoxy, ethanoyl and hydroxyl groups and protons which bind to C-4, C-4′ and C-3″. Interestingly, these bioactive compounds showed various medicinal properties based on our investigation on antioxidant, anticholinesterase and anti-HIV-1 protease activities. The IC_50_ value of DPPH and ABTS (antioxidant activities) was in the following descending order: polygonumins B > polygonumins C > polygonumins A > polygonumins D. In addition, almost similar pattern of antioxidant activity was observed for anti-acetylcholinesterase activity based on its IC_50_ value in descending order: polygonumins B > polygonumins C > polygonumins D > polygonumins A. On the other hand, polygonumins C and D showed inhibition of HIV-1 protease activity more than the positive control, pepstatin A. Finally, molecular docking studies on AChE and BChE proteins were carried out in order to gain insight into the mode of interactions between these compounds and the active residues for both enzymes. These remarkable findings indicate that these compounds have potential to be developed as targeted drugs for Alzheimer’s disease or as anti-HIV drugs.

## 1. Introduction

Due to its wealth of biodiversity, Malaysia is renowned for its herbal plant medicine, which has been used traditionally for more than a thousand years before the development of conventional or pharmaceutical drugs. Nowadays, herbal medicine has begun a “renaissance” era once again due to the side effects of synthetic drugs, reduced toxicity, the inability of modern therapies to cure several chronic diseases and microbial drug resistance. Moreover, traditional herbs are affordable and abundantly available, thus their usage is tremendously high compared to modern drugs. Large pharmaceutical companies, well-equipped with high-end technology, have begun to rediscover herbs as potential candidates for new natural product drug development, although they are facing great challenges in developing these drugs [1,2]. With recent advances, for example, new analytical and bioinformatics techniques, new strategies have arisen to hasten natural product drug discovery [3].

The passion for rediscovering the potential value of Malaysian herbs has led us to take this opportunity to highlight a well-known traditional herbal plant in Malaysia, *Polygonum minus* Huds. Polygonaceae, the synonym for *Persicaria minor*, is commonly referred to as ‘kesum’ or ‘laksa’ leaf in the Malay language [4,5]. The plant originates in southeast Asian countries, including Malaysia, Thailand, Vietnam and Indonesia, and is known to grow in the wild in damp areas near riverbanks, ditches and lakes. This plant has been found in Europe and Australia [6] and it is known as pygmy smartweed [7], small water pepper and swamp willow weed [8]. It is locally used as a flavoring ingredient in culinary dishes and may be consumed raw as ulam (Malaysian salad). Apart from that, this plant is traditionally used in herbal or alternative medicine as a cure for digestive disorders and dandruff [9]. *P. minus* produces a substantial amount of volatile compounds from the whole plant; its essential oil [10] can be used in aromatherapy and perfume manufacturing [11]. The constituents of kesum extracts correspond to the whole range of metabolites, such as polyoxygenated aromatics (polygonumins A), phenolic compounds, flavonoids, chalcones, anthraquinones, naphthoquinones, sesquiterpenoids, lignans, coumarins and stilbene glycosides [12,13,14]. These extracts have also been reported to exhibit several pharmacological properties such as antimicrobial activity [15,16,17], anti-atherosclerosis [18], antioxidant activity and anticancer activity [19]. Furthermore, kesum leaves are considered safe to consume up to 2000 mg/kg, as tested in Sprague Dawley rats [20].

Encouraged by the aforementioned findings, we aimed to isolate more new bioactive compounds from kesum as part of our continued investigations on kesum as a medicinal plant. In the present study, we isolated three new polygonumins derivatives, namely polygonumins B (1), C (2) and D (3), which were subjected to in vitro studies for antioxidant, anticholinesterase and anti-HIV protease activities. For anticholinesterase activity, we tested two enzymes, i.e., acetylcholinesterase (AChE) and butyrylcholinesterase (BChE), which play a vital role in co-regulating acetylcholine (ACh) levels in the brain. Herein, we present the discovery, structural elucidation and biological activities of these compounds. Molecular docking studies were carried out to further explore the possibility of the structural topographies required for the interaction of polygonumins derivatives with the BChE and AChE enzymes. This technique provides new insight into how the active compound model will bind to the target proteins and revealed the possible key active site residues involved in the intermolecular interactions. Nowadays, the molecular docking technique has become one of the most important processes for medicinal chemists in discovering their bioactive compounds (i.e., novel drugs) and even predict their activities before synthesis [21]. The molecular docking experiment has been done to support the anticholinesterase activity of certain bioactive compounds found in herbs [22,23].

## 2. Results and Discussion

Polygonumins A was successfully isolated and identified as reported previously and the compound is nearly identical to vanicoside A, isolated from *Polygonum pensylvanicum* with a few structural differences [24]. In vanicoside A, the substituents at C-4, C-2′ and C-4′ are hydroxyl, ethanoyl and hydroxyl units, respectively. However, the substituents at C-4, C-2′ and C-4′ in polygonumins A are ethanoyl, hydroxyl, and ethanoyl units, respectively. In addition, Carbon C-3″″′ of vanicoside A binds a methoxyl unit while polygonumins A binds a proton unit [25]. Polygonumins A is protected under Malaysia patent filling number AP 2014700594. The detailed structure of polygonumins A is supported by NMR data in Appendix A. Meanwhile in this study, our focus is on the newly isolated polygonumins derivatives polygonumins B, C and D.

Polygonumins B (1) is a white amorphous compound with m.p 150.0–151.0 °C. FTIR spectra indicated that the compound has various functional groups, i.e., hydroxyl absorbance at 3318.1 cm^−1^, aliphatic carbon absorbance at 2951.5 cm^−1^, carbonyl absorbance at 1689 cm^−1^ and aromatic absorbance at 1631, 1603 and 1512 cm^−1^. According to the HRESIMS data, polygonumins B (1) has a molecular weight of *m/z* 1040.2697 with molecular formula of C_53_H_52_O_22_. The results are supported by NMR data (1H and 13C NMR), which showed that the structure of polygonumins B (1) comprises four phenylpropanoid units and a sucrose unit, which could be well distinguished from the structure of polygonumins A. NMR data for polygonumins A and polygonumins B (1) are compared in Appendix A and Figure 1. Polygonumins A has a molecular formula of C_52_H_50_O_21_ and a molecular weight of 1010 mass units. The structural differences between polygonumins A and B (1) are located at C-3″. Polygonumins A binds a proton unit with a chemical shift δH of 6.94 (*d,* 8.4, 1 H), while polygonumins B (1) binds a methoxy group at δC 55.5 ppm.

Furthermore, structural differences between polygonumins B (1), C (2) and D (3) were found at C-4, C-4′ and C-3″, where B (1) had two ethanoyls bound to C-4 and C4′ and also had a methoxy bound at C-3″. In polygonumins C (2), the substituents at C-4, C-4′ and C-3″ are ethanoyl, hydroxyl and proton units, respectively. For polygonumins D (3), it had a proton at C-3″ and two hydroxyls were bound to C-4 and C-4′. The substituents at C-4, C-2′ and C-4′ in polygonumins A are ethanoyl, hydroxyl and ethanoyl units, respectively. In the HMBC spectrum of polygonumins A in particular, the first ethanoyl unit shows long-range 1H-13C correlations between the signal δH 4.98 ppm (H-4′) and the carbonyl signal δC 170.1 ppm (C-2′″″″) and between the signal δH 1.99 ppm (H-1″″″′) and the carbonyl signal δC 170.1 ppm (C-2′″″″). The second ethanoyl unit of polygonumins A shows long-range 1H-13C correlations between the signal δH 4.83 ppm (H-4) and the carbonyl signal δC 169.6 ppm (C-2″″″) and between the signal δH 1.99 ppm (H-1″″″′) and the carbonyl signal δC 169.6 ppm (C-2″″″) (Figure 1). NMR data of Polygonumins A, B, C and D were compared in Appendix A).

All polygonumins derivatives and vanicosides are phenylpropanoid sucrose esters, which are found naturally in plants, especially in *polygonum* spp., and this group is proven to have many biological activities [26]. For example, vanicoside B has antitumor activities against breast cancer [27].The structure of vanicosides A and B can be well distinguished from polygonumins derivatives (except polygonumins B) due to its methoxyl group located at C-3″. On the other hand, polygonumins A, B and C are different from vanicosides A and B because of the ethanoyl group located at C-4 on furanose ring (Figure 1).

Polygonumins A and B isolated from *Polygonum orientale* L. were reported in 2001 [28]. This polygonumin belongs to limonoid group and is different from our previously reported polygonumins A. In *Polygonum orientale* L., polygonumins A was assigned as trans-feruloyldeacetylnomilin; on the other hand, polygonumins B was assigned as cis-ferulolyldeacetylnomilin. Polygonumins A–D reported here belong to phenylpropanoid sucrose esters, which are similar to vanicosides’ basic structure.

The antioxidant activity of the isolated compounds was studied by free radical inhibition (DPPH and ABTS) compared to gallic acid as the reference compound (Table 1). Polygonumins B showed very high antioxidant activity (DPPH assay), with an IC_50_ of 27 μg/mL, followed by gallic acid, polygonumins C, polygonumins A and polygonumins D. In this study, polygonumins B had the highest antioxidant activity due to the existence of a methoxyl group (OCH3) at the C-3″ position compared to the positive control, gallic acid and the other derivatives that had no methoxyl group. At this point, we can conclude that the methoxyl group in these derivatives is the antioxidant activity center. According to a reported study on antioxidant activity [29] the greater the number of methoxyl groups, the higher the antioxidant activity with the same mother nucleus structure of the phenolic group. The presence of a methoxyl group also reduced the ionization potential (IP) value, which enhanced the activity. For ABTS activity, gallic acid had the highest scavenging activity, followed by polygonumins B, C, A and D. Although their rankings differed slightly, polygonumins B still had the highest antioxidant capacity among the other polygonumins derivatives, explained by the existence of a methoxyl group as the antioxidant center. Polygonumins C, which showed the second highest DPPH antioxidant activity, with an IC_50_ of 240 µg/mL, on the other hand did not have a methoxyl group at the C-3″ position. The antioxidant activity was due to the presence of an acetyl group at C-4′ and the position of a hydroxyl group at C-4′. Compared to polygonumins A, polygonumins C has more hydroxyl groups, which increases the free radical scavenging activity. This result has been supported by previous studies on stilbene [30] which showed that the radical scavenging activity depends on the number and position of hydroxyl groups, i.e., the activity is increased significantly with a greater number of hydroxyl groups. For ABTS scavenging activity, gallic acid had the highest antioxidant capacity due to its three hydroxyl groups bound to an aromatic ring. Furthermore, the ABTS assay favors compounds that contain hydrophilic (high solubility) and lipophilic structures [31] such as gallic acid, compared to the DPPH assay in reflecting antioxidant capacity. Even though both the DPPH and ABTS assays have their hurdles and pitfalls, a similar pattern of antioxidant activity among polygonumins derivatives was observed in both assays: polygonumins B > polygonumins C > polygonumins A > polygonumins D. Our study revealed that the contribution of hydroxyl groups to antioxidant activity should not be ruled out, as some research has shown that the absence of a hydroxyl group will result in no antioxidant activity [32]. Apart from methoxyl and hydroxyl groups, we hypothesized that the acetyl moiety that is attached at C-4 (fructose) also plays a vital role in enhancing the antioxidant activity in polygonumins B and polygonumins C. This is supported by a report showing that the presence of acetyl groups can increase the antioxidant activity in polysaccharides, as acetylated polysaccharides are able to donate more electrons by reducing hydrogen bonds and activating hydrogen on anomeric carbon, which quickly terminates the radical chain reaction [33,34].

Acetylcholine (ACh) is a neurotransmitter inhibited by two enzymes: acetylcholinesterase (AChE) and butyrylcholinesterase (BChE). AChE is found in high concentrations, mainly in red blood cells as well as in the nervous system. The enzyme BChE (also known as “pseudo” cholinesterase; EC 3.1.1.8) is a non-specific type of cholinesterase enzyme that hydrolyzes different types of choline esters. It exists throughout the body, most importantly in the human liver, blood serum, pancreas and central nervous system. In the brain, BChE is primarily associated with glial cells and endothelial cells [35]. In Alzheimer’s disease (AD), both AChE and BChE are found at high levels, and both enzymes play a vital role in coregulating the level of Ach [36], although it was suggested at first that AChE predominates in the healthy brain, while BChE plays a minor role in regulating ACh levels in the brain. Reports have shown that, as AD progresses, the level of BChE increases, leading to neuronal degeneration [37]. Inhibition of BChE not only increases the level of ACh in the brain, but also impedes the formation of beta-amyloid plaques, which contribute to AD [38]. Owing to its vital role in regulating the ACh level, we decided to test our compounds with BChE as well. A similar pattern to antioxidant activity was observed for anti-acetylcholinesterase activity, whereby polygonumins B showed high anti-acetylcholinesterase activity (71.81%), followed by polygonumins C (59.15%). The AChE and BChE inhibitory activities of all the compounds are summarized in Table 2. This activity occurred due to the meta position of methoxyl at the feruloyl moiety, which increased the anticholinesterase activity. In addition, there is a report showing that the addition of an electron donor group, i.e., a methoxy on the phenyl ring at the meta position, leads to higher anticholinesterase activity [39]. For anti-butyrylcholinesterase activity, the overall inhibition activity was not that high when compared to anti-acetylcholinesterase, where polygonumins C showed the highest activity among the other compounds, with 42.71% inhibition. This may suggest that these compounds might be interacting with the enzymes by different mechanisms. Studies have also shown that high levels of antioxidants may also decrease the incidence and prevalence of AD [40]. With these findings, we can conclude that polygonumins B and C, which have high antioxidant activity and showed inhibitory activity against both AChE and BChE, have the potential to be developed as a treatment for AD. Moreover, the present drugs tacrin, rivastigmin and donepezil with AChE inhibitory activity have some side effects and are effective only against the mild type of AD. There are no drugs available with BChE inhibitory activity at present [41].

Investigations into the anti-HIV protease activity of these compounds were carried out as previously; polygonumins A showed moderate inhibitory activity [25]. Of the tested compounds, polygonumins C and polygonumins D showed the highest inhibitory activity, more than the positive control (pepstatin A), both with 91% inhibition at 100 µg/mL (Table 3). However, polygonumins B showed weak anti-HIV activity compared to the other compounds, although this compound showed high activity against AChE and BChE.

In previous reports, the phenyl propanoid glucoside moiety in polygonumins A and vanicoside A was associated with anti-HIV protease activity [25,42]. However, any functional group that exists, absent, or substitutes with the other functional group in the phenylpropanoid glucoside moiety also plays a critical role in the determination of protease activity. In the case of both polygonumins C and polygonumins D, the absence of an ethanoyl unit at C-4′, and the absence of proton and methoxy units at C-3″, seem to have enhanced the anti-HIV protease activity. It seems that the methoxy group at C-3″ in polygonumins B further decreased the anti-protease activity. On the other hand, the substitution of an ethanoyl group at C-4 with a hydroxyl group in polygonumins C and D does not mediate this activity. Our findings are supported by a previous report showing that a slight chemical structure difference at a sugar unit (glycoside), for example, substitution of a methyl group with a hydroxyl methyl group, led to huge differences in the anti-HIV activity [43]. This structural finding is very important for future drug design and synthesis studies.

To gain insight into the specific binding of the isolated compounds with BChE and AChE, molecular docking analysis was performed and 2D interactions between the compounds and the proteins were observed. Well-defined binding sites have been demonstrated in the published crystal structures of both BChE and AChE and were reported to conserve key interactions, namely Pi–Pi with Trp82/Trp86 and hydrogen bonding with His438/His447 in BChE/AChE, respectively [44]. In addition to the conserved residues in both proteins, some other common interactions have been reported, including Asp70/Asp74, Ser198/Ser203, Ala328/Tyr337, Trp430/Trp439, Met434/Met443, and Met437/Pro446 of BChE/AChE, respectively [45]. To sum up, the main residues in the active sites of BChE and AChE have been classified into four groups: peripheral binding sites, catalytic residues, the acyl-binding pocket and catalytic anionic subsites [46,47]. The details of the main active residues are highlighted in Figure 2a for BChE and Figure 2b for AChE.

The docking results for BChE revealed that polygonumins B, C and D were stably positioned in the catalytic domain of BChE residues, 5 Å from the inhibitor of the original complex with the lowest binding energy compared to polygonumins A. During the docking process, the ligands were fully flexible, the protein was rigid, and the binding site was defined as a sphere with a radius of 5 Å. In addition, the binding energies of polygonumins B, C and D were −9.4, −9.5 and −9.2, respectively, indicating high affinities of BChE residues (Table 4). The superimposed structure of the polygonumins derivatives on the BChE protein are illustrated in Figure 3a. The hydrophobicity surface view of polygonumins derivatives on BChE residues is in Appendix A).

From our observation, in all three compounds that exhibit the lowest binding energies and the highest BChE inhibition activity, two of the benzene rings are specifically engaged in T-stacking (-interaction) with Trp82 and Ala328 in the catalytic anionic subsite. The oxygen from the carbonyl group and the aromatic ring of these three compounds are attached to the hydrogens of the acyl-binding pocket (Gly 116 and Gly 117) to form hydrogen interactions. The isolated compounds with high BChE activity were found to interact at four or more active site residues. The detailed binding interactions between all these compounds and BChE are shown in Appendix A). The biological assays confirmed that polygonumins C was the most potent BChE inhibitor, followed by polygonumins D and polygonumins B. The interactions of these compounds with the catalytic anionic subsite site and the acyl-binding pocket play a vital role in BChE inhibition.

On the other hand, the molecular docking analysis of AChE showed that polygonumins B and C had the best interactions with the enzyme crystal structure, with scores of −9.2 and −9.4, respectively (Table 5). The superimposed structure of the polygonumins derivatives on the AChE protein are illustrated in Figure 3b. The results for both compounds were counter-verified using in vitro studies (inhibition percentage of the enzyme activity), showing that polygonumins B had the highest inhibition activity, followed by polygonumins C. Tacrine and polygonumins B both had two of their benzene rings T-stacking (-interaction) to Phe330 in the catalytic anionic subsite, whereas polygonumins C only has one benzene ring that attaches to Phe330. Tacrine also has two of its benzene rings engaged hydrophobically with Trp84, whereas in polygonumins B, one benzene ring showed two types of interaction with Trp84: hydrogen interactions that make contact through oxygen molecules that bind to the benzene ring and hydrophobic interactions of the benzene ring. Both polygonumins B and C, which showed the lowest binding energy, were engaged through similar established AChE active gorge residues at the peripheral anionic site (Trp279, Tyr70 and Ser286), the catalytic anionic subsite (Trp84 and Phe330) and at the acyl-binding pocket (Gly118). According to our model, the high inhibition activity of this compound is due to interactions with the peripheral anionic site (PAS), catalytic anionic subsite (CAS) and acyl-binding pocket (ABP) of AChE as the compounds are engaged with many residues from these groups. The presence of interactions with PAS, CAS and ABP seems compulsory for the recognition of polygonumins compounds as AChE inhibitors. The detailed binding interactions between these compounds and AChE are shown in Appendix A).

Overall, the AChE– and BChE–ligand complex docking results revealed that the polygonumins derivatives were stably positioned in several pockets/catalytic regions of AChE and BChE, all of which were 5 Å from the inhibitor in the original complex (Figure 3a,b). The data clearly show that polygonumins derivatives prefer binding at hydrophilic protein pockets in the BChE; however, the polygonumins derivatives prefer binding at hydrophobic protein pockets in the AChE, as can be observed in Appendix A.

## 3. Conclusions

In summary, we have successfully isolated polygonumins derivatives (polygonumins B, C and D) from the stems of *Polygonum minus* and these compounds have potent anticholinesterase activities and anti-HIV protease activities. Polygonumins B and polygonumins C showed promising anticholinesterase activities among all the polygonumins derivatives. Moreover, the molecular docking analysis and anticholinesterase in vitro assay were discovered to be in agreement. This suggests that polygonumins B and polygonumins C have the potential to be developed as a treatment of AD in the future. Our investigation also revealed that polygonumins C and polygonumins D showed high HIV protease inhibition activity compared to positive control, suggesting both are suitable therapeutic candidates for anti-HIV drugs. Among the single compounds isolated, polygonumins C is worth investigation as it has the most medicinal properties and showed both anticholinesterase and anti-HIV protease activities.

## 4. Materials and Methods

### 4.1. Plant Materials

The whole parts of *P*. *minus* were originally collected between December 2017 to January 2018 from an INBIOSIS experimental plot, Universiti Kebangsaan Malaysia, and a voucher specimen was deposited in the UKMB Herbarium, Universiti Kebangsaan Malaysia (PM-2014-1). Specimens were identified by a taxonomist and further confirmed by ITS sequencing [49]. The fresh samples were cleared and washed using running tap water then dried using an oven (Memmert UF110, Memmert Universal, Schwabach, Germany)) at 40 °C and then ground for approximately 2–3 min using a grinder (Multifunction disintegrator SY-04, Golden Bull). 

### 4.2. Chemicals and Reagents

The chemicals used for the extraction process were methanol (99.9% purity, Merck, Darmstadt, Germany), *n* hexane (99% purity, Sigma-Aldrich, Darmstadt, Germany), distilled water, ethanol (95% purity, Merck, Darmstadt, Germany), acetone (Sigma-Aldrich, Darmstadt, Germany), dichloromethane (Sigma-Aldrich, Darmstadt, Germany), and carbon dioxide (99.9% purity, Alpha Gas Solution, Selangor, Malaysia). For the bioassay tests, the chemicals and reagents used were gallic acid (99% purity, Merck, Darmstadt, Germany), 1,1-diphenyl-2-picrylhydrazyl (≥85% purity, Sigma-Aldrich, Burlington, MA, USA), 2,2′-azino-bis(3-ethylbenzothiazoline-6-sulfonic acid) diammonium salt (≥98% purity, Sigma-Aldrich, Burlington, MA, USA), methanol and ethanol (99.9% purity, Merck, Darmstadt, Germany). Acetylthiocholine iodide (ATCI), AChE, bovine serum albumin (BSA), 5,5-Dithio(bis)nitrobenzoic acid (DTNB) and galanthamine were obtained from Sigma-Aldrich, Steinheim, Germany). The AChE used in the assay was from electric eel (type-VI-S, EC 3.1.1.7, Sigma-Aldrich, St. Louis, MO, USA), BChE was from horse (EC 3.1.1.8) and butyrylthiocholin chloride was purchased from Sigma (St. Louis, MO, USA). The HIV-1 protease inhibitor screening kit (fluorometric) was purchased from Biovision Incorporated (Milpitas, CA, USA). All chemicals and reagents used in the study were of analytical grade.

### 4.3. Compound Identification

The structures of the purified compounds were determined based on spectral data recorded on a Frontier Perkin-Elmer FTIR/NIR (Perkin-Elmer Inc., Norwalk, CT, USA) spectrophotometer and a Bruker NMR 600 MHz and 700 MHz Cryo-Probe instrument that could perform 1-D and 2-D NMR measurements (Bruker, Berlin, Germany). ESIMS were recorded on a Bruker Daltonics micrOTOF-Q III (positive polarity). Isolation was carried out by radial chromatography using round glass plates on a Merck Kieselgel 60 PF_254_ (art. no. 1.07749.1000) and the profile was analyzed using aluminum sheets measuring 20 × 20 cm on a Merck TLC silica gel 60 F_254_ with a thickness of 0.25 mm (art. no. 1.05554.0001) with UVP light detection at 254 nm (UVGL-58 Handheld UV Lamp, USA) or CeSO_4_ spraying, followed by heating.

### 4.4. Isolation of Bioactive Compounds

Dried ground stem bark of *Polygonum minus* (5 kg) was macerated with 3 × 4 L of methanol by agitation at room temperature. The extracts were combined and concentrated using a rotary evaporator at 50 °C to yield 120 g of dark green extract. The whole extract was fractionated on a silica gel column eluted with a gradient of increasing polarity of chloroform to 100% methanol. The eluents were monitored by thin layer chromatography (TLC) and combined to give five fractions. Fraction four was further subjected to column chromatography (D = 5 cm) using a varied ratio of chloroform and methanol as the solvent system. The eluates showing the same profile on TLC were combined to give four subfractions. Purification of subfraction 4.2–4.3 (3.1 g) was carried out using radial chromatography (RC) with a silica gel plate of 2 mm thickness eluted with chloroform and methanol (8.5:1.5) in 5% polarity increments to yield polygonumins A (107 mg) and polygonumins B (34 mg). Purification of subfraction 4.4 (3.3 g) was conducted by utilizing another RC with a silica gel plate of 2 mm thickness. Elution with chloroform and methanol (7:3) produced polygonumins C (20 mg) and polygonumins D (15 mg). A schematic flow chart of the isolation and extraction procedures is shown in Figure 4.

#### 4.4.1. Polygonumins B (1)

White amorphous powder; m.p 150.0–151.0 °C; HRESIMS [M-H]^−^ at *m/z* 1039.2697 (1039.9642 ([M-H]^−^ calc.) or [M] at *m/z* 1040.2697 (1040.9642 ([M] calc.); FTIR spectra showed absorption bands at (cm^−1^) 3334.8 (hydroxyl); 2953.3 (C-H aliphatic); 1704 (C-carbonyl) and 1629.4; 1605.3, and 1513.7 (aromatics rings); ^1^H NMR (700 MHz; acetone-*d*_6_): *fructose*-δ_H_ 4.38 (1H, *d*, 4.4, H-1a), 4.20 (1H, *d*, 12.0, H-1b), 5.62 (1H, *d*, 7.9, H-3), 4.82 (1H, *dd*, 3.7, 10.8, H-4), 4.30 (1H, *t*, 8.0, H-5), 4.62 (1H, *dd*, 12.1, 3.4, H-6a), 4.48 (1H, *d*, 6.8, H-6b); *glucose*-δ_H_ 5.76 (1H, *d*, 3.7, H-1′), 4.68 (1H, *t*, 8.0, H-2′), 4.50 (1H, *t*, 7.1, H-3′), 4.96 (1H, *t*, 9.4, H-4′), 4.11 (1H, *q*, 18.0, 9.6, H-5′), 4.41 (1H, *dd*, 2.2, 12.2, H-6′a), 4.21 (1H, *d*, 4.9, H-6′b); *feruloyl moiety*-δ_H_ 7.38 (1H, *s*, H-2″), 6.90 (1H, *d*, 8.4, H-5″), 7.15 (1H, *d*, 8.2, H-6″), 7.78 (1H, *d*, 15.8, H-7″), 6.54 (1H, *d*, 15.9, H-8″), 3.91 (3H, *s*, 3″-OCH_3_); *p-coumaroyl moieties*-δ_H_ 7.54 (2H, *d*, 8.64, H-2/6′″), 7.56 (2H, *d*, 8.64, H-2/6″″), 7.56 (2H, *d*, 8.64, H-2/6′″″), 6.89 (2H, *d*, 8.4, H-3/5′″), 6.89 (2H, *d*, 8.4, H-3/5″″), 6.94 (2H, *d*, 8.52, H-3/5′″″), 7.71 (1H, *d*, 15.96, H-7′″), 7.67 (1H, *d*, 9.18, H-7″″), 7.65 (1H, *dd*, 8.28, 1.74, H-7′″″), 6.51 (1H, *d*, 15.9, H-8′″), 6.45 (1H, *d*, 18.0, H-8″″), 6.40 (1H, *d*, 15.96, H-8′″″); *ethanoyl moieties*-δ_H_ 2.0 (3H, *s*, H-2″″″), 2.1 (3H, *s*, H-2″″″′). ^13^C NMR (175 MHz; acetone-*d*_6_): *fructose*-δ_C_ 65.0 (C-1), 102.6 (C-2), 78.2 (C-3), 72.6 (C-4), 80.3 (C-5), 64.3 (C-6); *glucose*-δ_C_ 89.2 (C-1′), 73.3 (C-2′), 68.6 (C-3′), 71.4 (C-4′), 68.8 (C-5′), 63.1 (C-6′); *feruloyl moiety*-δ_C_ 126.5 (C-1″), 110.4 (C-2″), 149.3 (C-3″), 147.9 (C-4″), 116.0 (C-5″), 123.4 (C-6″), 146.1 (C-7″), 115.1 (C-8″), 167.0 (C-9″), 55.5 (C-3″OCH_3_); *p-coumaroyl moieties*-δ_C_ 126.0 (C-1′″), 125.96 (C-1″″), 125.9 (C-1′″″), 130.2 (C-2/6′″), 130.23 (C-2/6″″), 130.4 (C-2/6′″″), 115.84 (C-3/5′″), 115.84 (C-3/5″″), 116.0 (C-3/5′″″), 159.9 (C-4′″), 159.92 (C-4″″), 160.2 (C-4′″″), 145.5 (C-7′″), 145.3 (C-7″″), 145.1 (C-7′″″), 114.1 (C-8′″), 114.2 (C-8″″), 114.5 (C-8′″″), 166.0 (C-9′″), 166.03 (C-9″″), 166.5 (C-9′″″); *ethanoyl moieties*-δ_C_ 170.1 (C-1″″″), 20.0 (C-2″″″′), 169.6 (C-1″″″′), 20.1 (C-1″″″′). (For detailed assignment, see Appendix A).

#### 4.4.2. Polygonumins C (2)

White amorphous; m.p 156.0–157.0 °C; HRESIMS [M-H]^−^ at *m/z* 967.2697 (967.9642 ([M-H]^−^ calc.) or [M] at *m/z* 968.2697 (968.9642 ([M] calc.); FTIR spectra showed absorption bands at (cm^−1^) 3334.8 (hydroxyl); 2953.3 (C-H aliphatic); 1704 (C-carbonyl) and 1629.4; 1605.3, and 1513.7 (aromatics rings); ^1^H NMR (700 MHz; acetone-*d_6_*): *fructose*-δ_H_ 4.20 (1H, *d*, 11.6, H-1a), 4.33 (1H, *d*, 11.6, H-1b), 5.63 (1H, *d*, 8.3, H-3), 4.68 (1H, *t*, 11.6, H-4), 4.30 (1H, *q*, 11.0, H-5), 4.55 (1H, *d*, 15.4, H-6a), 4.61 (1H, *d*, 12.1, H-6b); *glucose*-δ_H_ 5.70 (1H, *d*, 3.5, H-1′), 4.73 (1H, *t*, 4.6, H-2′), 3.96 (1H, *t*, 4.7, H-3′), 4.52 (1H, *br*-*t*, 5.1, H-4′), 4.34 (1H, *m*, H-5′), 4.30 (1H, *d*, 11.1, H-6′a), 4.59 (1H, *d*, 15.4, H-6′b); *feruloyl moiety*-δ_H_ 7.55 (1H, *d*, 8.8, H-2″), 6.93 (1H, *m*, H-3″), 6.90 (1H, *d*, 8.4, H-5″), 7.56 (1H, *d*, 8.8, H-6″), 7.76 (1H, *d*, 16.0, H-7″), 6.52 (1H, *d*, 16.0, H-8″); *p-coumaroyl moieties*-δ_H_ 7.62 (2H, *d*, 8.3, H-2/6′″), 7.58 (2H, *d*, 7.86, H-2/6″″), 7.56 (2H, *d*, 8.8, 2/6′″″), 6.87 (2H, *m*, 3/5′″), 6.90 (2H, *m*, 3/5″″), 6.93 (2H, *m*, 3/5′″″), 7.73 (1H, *d*, 16.3, H-7′″), 7.67 (1H, *d*, 16.6, H-7″″), 7.65 (1H, *d*, 12.8, H-7′″″), 6.49 (1H, *d*, 16.0, H-8′″), 6.46 (1H, *d*, 16.1, H-8″″), 6.40 (1H, *d*, 16.0, H-8′″″); *ethanoyl moieties*-δ_H_ 2.1 (3H, *s*, H-2″″″). ^13^C NMR (175 MHz; acetone-*d_6_)*: *fructose*-δ_C_ 64.5 (C-1), 102.4 (C-2), 77.9 (C-3), 72.9 (C-4), 80.1 (C-5), 63.93 (C-6); *glucose*-δ_C_ 89.44 (C-1′), 77.7 (C-2′), 68.9 (C-3′), 73.1 (C-4′), 71.0 (C-5′), 64.65 (C-6′); *feruloyl moiety*-δ_C_ 126.2 (C-1″), 130.21 (C-2″), 115.8 (C-3″), 159.72 (C-4″), 115.8 (C-5″), 130.21 (C-6″), 146.03 (C-7″), 113.7 (C-8″), 166.03 (C-9″); *p-coumaroyl moieties*-δ_C_ 126.1 (C-1′″), 126.0 (C-1″″), 125.9 (C-1′″″), 130.24 (C-2/6′″), 130.25 (C-2/6″″), 130.5 (C-2/6′″″), 115.76 (C-3/5′″), 115.83 (C-3/5″″), 115.88 (C-3/5′″″), 159.78 (C-4′″), 159.86 (C-4″″), 160.0 (C-4′″″), 145.23 (C-7′″), 145.03 (C-7″″), 144.9 (C-7′″″), 114.2 (C-8′″), 114.3 (C-8″″), 114.6 (C-8′″″), 166.11 (C-9′″), 166.5 (C-9″″), 166.8 (C-9′″″); *ethanoyl moieties*-δ_C_ 170.2 (C-1″″″), 20.17 (C-2″″″′).(For detailed assignments, see Appendix A).

#### 4.4.3. Polygonumins D (3)

White amorphous; m.p 156.0–157.0 °C; HRESIMS [M-H]^−^ at *m/z* 925.2697 (925.9642 ([M-H]^−^ calc.) or [M] at *m/z* 926.2697 (926.9642 ([M] calc.); FTIR spectra showed absorption bands at (cm^−1^) 3334.8 (hydroxyl); 2953.3 (C-H aliphatic); 1704 (C-carbonyl) and 1629.4; 1605.3, and 1513.7 (aromatics rings); ^1^H NMR (700 MHz; acetone-*d_6_*): *fructose*-δ_H_ 4.20 (1H, *d*, 11.6, H-1a), 4.33 (1H, *d*, 11.6, H-1b), 5.63 (1H, *d*, 8.3, H-3), 4.68 (1H, *t*, 11.6, H-4), 4.30 (1H, *q*, 11.0, H-5), 4.55 (1H, *d*, 15.4, H-6a), 4.61 (1H, *d*, 12.1, H-6b); *glucose*-δ_H_ 5.70 (1H, *d*, 3.5, H-1′), 4.73 (1H, *t*, 4.6, H-2′), 3.96 (1H, *t*, 4.7, H-3′), 4.52 (1H, *br*-*t*, 5.1, H-4′), 4.34 (1H, *m*, H-5′), 4.30 (1H, *d*, 11.1, H-6′a), 4.59 (1H, *d*, 15.4, H-6′b); *feruloyl moiety*-δ_H_ 7.55 (1H, *d*, 8.8, H-2″), 6.93 (1H, *m*, H-3″), 6.90 (1H, *d*, 8.4, H-5″), 7.56 (1H, *d*, 8.8, H-6″), 7.76 (1H, *d*, 16.0, H-7″), 6.52 (1H, *d*, 16.0, H-8″); *p-coumaroyl moieties*-δ_H_ 7.62 (2H, *d*, 8.3, H-2/6′″), 7.58 (2H, *d*, 7.86, H-2/6″″), 7.56 (2H, *d*, 8.8, 2/6′″″), 6.87 (2H, *m*, 3/5′″), 6.90 (2H, *m*, 3/5″″), 6.93 (2H, *m*, 3/5′″″), 7.73 (1H, *d*, 16.3, H-7′″), 7.67 (1H, *d*, 16.6, H-7″″), 7.65 (1H, *d*, 12.8, H-7′″″), 6.49 (1H, *d*, 16.0, H-8′″), 6.46 (1H, *d*, 16.1, H-8″″), 6.40 (1H, *d*, 16.0, H-8′″″); *ethanoyl moieties*-δ_H_ 2.1 (3H, *s*, H-2″″″). ^13^C NMR (175 MHz; acetone-*d_6_*): *fructose*-δ_C_ 64.5 (C-1), 102.4 (C-2), 77.9 (C-3), 72.9 (C-4), 80.1 (C-5), 63.93 (C-6); *glucose*-δ_C_ 89.44 (C-1′), 77.7 (C-2′), 68.9 (C-3′), 73.1 (C-4′), 71.0 (C-5′), 64.65 (C-6′); *feruloyl moiety*-δ_C_ 126.2 (C-1″), 130.21 (C-2″), 115.8 (C-3″), 159.72 (C-4″), 115.8 (C-5″), 130.21 (C-6″), 146.03 (C-7″), 113.7 (C-8″), 166.03 (C-9″); *p-coumaroyl moieties*-δ_C_ 126.1 (C-1′″), 126.0 (C-1″″), 125.9 (C-1′″″), 130.24 (C-2/6′″), 130.25 (C-2/6″″), 130.5 (C-2/6′″″), 115.76 (C-3/5′″), 115.83 (C-3/5″″), 115.88 (C-3/5′″″), 159.78 (C-4′″), 159.86 (C-4″″), 160.0 (C-4′″″), 145.23 (C-7′″), 145.03 (C-7″″), 144.9 (C-7′″″), 114.2 (C-8′″), 114.3 (C-8″″), 114.6 (C-8′″″), 166.11 (C-9′″), 166.5 (C-9″″), 166.8 (C-9′″″); *ethanoyl moieties*-δ_C_ 170.2 (C-1″″″), 20.17 (C-2″″″′). (For detailed assignments, see Appendix A).

### 4.5. DPPH Free Radical Scavenging Assay

Radical scavenging activities were determined by the DPPH (2,2-diphenyl-1-picrylhydrazyl) assay with some modifications [50]. Various concentrations of the experimental samples were taken, and the volume was adjusted to 200 μL with ethanol. About 100 μL of 0.5 mM DPPH in methanol was added to the samples (or standard gallic acid) in a 96-well plate. After incubation for 30 min in the dark, changes in absorbance at 600 nm were measured on a 96-well plate reader. A DPPH control was prepared without the compound. The percentage of inhibition was calculated as below:DPPH scavenging activity: (Ac − As)/Ac ∗ 100 
where Ac is the absorbance of the control (DPPH solution without sample) and As is the absorbance of the test sample with DPPH solution. Each sample was measured in triplicate. The IC_50_ was calculated as the sample concentration required for a 50% decrease in the absorbance of a control solution of DPPH.

### 4.6. ABTS Radical Scavenging Activity

The ABTS scavenging activities of the isolated compounds were performed by using a previous method with slight variations [51]. Briefly, the ABTS^•+^ reaction was initiated by mixing ABTS solution (7 mM) with potassium persulphate (2.45 mM) for 12–16 h in the dark at room temperature. Then, the ABTS solution was further diluted with methanol to achieve absorbance at 0.7 ± 4 nm at 734 nm and equilibrated for 15 min at room temperature. The methanolic solution of each compound of various concentrations (0.1 mL) was mixed with the previous ABTS diluted solution (0.9 mL) and incubated at room temperature for 7 min. The solution was read at 734 nm with gallic acid as the positive control. The percentage of inhibition was calculated as below:ABTS scavenging activity: (Ac − As)/Ac ∗ 100
where Ac is the absorbance of the control (ABTS solution without sample) and As is the absorbance of the test sample with ABTS solution. The IC_50_ was calculated as the sample concentration required for a 50% decrease in the absorbance of a control solution of ABTS.

### 4.7. Anti-Acetylcholinesterase (AChE) Activity and Anti-Butyrylcholinesterase Activity (BChE)

Inhibition of acetylcholinesterase (AChE) and butyrylcholinesterase (BChE) was assessed using the spectrophotometric method developed by Ellman [52] with slight modifications. Electric eel AChE (electric eel acetylcholinesterase, type-VI-S, EC 3.1.1.7, Sigma-Aldrich, St. Louis, MO, USA) and acetylthiocholine iodide (Sigma-Aldrich, Steinheim, Germany) were used as the enzyme and substrate, respectively. For BChE, equine serum (EC 3.1.1.8) was used as the enzyme and butyryl thiocholine iodide as the substrate. Briefly, 125 μL of DTNB (Sigma-Aldrich, Steinheim, Germany) (50 mM Tris-HCl, pH 8, 0.1 M NaCl, 0.02 M MgCl_2_·6H_2_O), 25 μL of AChE (0.2 U/mL) or BChE (0.2 U/mL), 25 μL of test compound solution in DMSO, and 50 μL of buffer (50 mM Tris-HCl, pH 8, 0.1% BSA) were mixed and incubated at 25 °C for 30 min. DMSO or buffer (25 μL) was added instead of the test compound solution in control experiments. The reaction was then initiated by the addition of 25 μL of acetylthiocholine iodide or butyryl thiocholine iodide (0.25 mmol/L), which brought the final volume to 250 μL. The formation of 5-thio-2-nitrobenzoate anion from the enzymatic hydrolysis of acetylthiocholine iodide was monitored based on the absorbance at 412 nm on a 96-well microplate reader (Model 680, BioRad Inc., Hercules, CA, USA). The reaction rates were calculated from data collected at specific time points over the first 180 s in 20 s increments. Percent inhibition of AChE and BChE were determined by the ratio of the reaction rate with the test sample to that with the blank control (DMSO in Tris-HCl buffer, pH 8.0) using the formula (E − S)/E × 100, where E is the activity of enzyme with DMSO and S is the activity of enzyme with the test sample. The experiments were carried out in triplicate. Tacrine was used as a reference compound.

### 4.8. HIV-1 Protease Fluorogenic Assay

In this study, the fluorometric enzymatic activity assay, which is a well-known method for measuring the inhibition of HIV-1 protease activity, was used [53]. HIV-1 protease inhibitory activity in the samples was determined by using an HIV-1 Protease Inhibitor Screening Kit (fluorometric) from Biovision Incorporated (Milpitas, CA, USA). Pepstatin A, aspartic acid protease (1 mM) was used as a known standard for HIV-1 protease inhibition and 1% DMSO was used as a solvent control, respectively. The assay was freshly prepared and performed according to the manufacturer’s instructions by using black-flat bottom 96-well microplates. Briefly, the sample (100 µg/mL) was incubated with HIV-1 protease at room temperature for 15 min. The fluorescence intensity of HIV-1 protease activity was measured at an excitation (Ex) wavelength of 330 nm and emission (Em) wavelength of 450 nm in one-minute intervals for 60 min at 37 °C using a Thermo Scientific Varioskan Flash (Thermo Fisher scientific, Vantaa, Finland). The reaction was started by the addition of 10 µL substrate as described in the kit instruction’s manual.

### 4.9. Molecular Docking Studies

The studies were run with the following parameters: 11th Gen Intel(R) Core (TM) i5-1135G7 @ 2.40 GHz/1.38 GHz; 8 GB RAM; Windows 10 operating platform. Protein–ligand docking was carried out using AutoDock Vina software. X-ray crystal structures, in complex with the inhibitors, of human BChE (PDB: 4bds with 2.10 Å resolution) and AChE (PDB: 2ckm with 2.15 Å resolution) from the RCSB protein data bank were selected as the target proteins based on suitable resolution and co-crystallized ligands, i.e., AChE and BChE. The 3D structures of the test compounds were constructed using molview software. The energy of ligands was minimized using the MMFF94x force field via Avogadro software. This optimized ligand was then used for docking studies with the AChE and BChE enzymes. AutoDock Vina 1.0 beta 0.2 software was used to search for potential targets of the compounds. The graphical user interface program AutoDock Tools was used to prepare, run and analyze the docking simulations. Atom charges, solvation parameters and polar hydrogens were added to the receptor PDB file for the preparation of the protein in the docking simulations. AutoDock Vina requires precalculated grid maps. This grid must contain the potential binding region of the target of interest. The grid box size was set at 20, 20 and 20 Å (x, y and z). The spacing between grid points was 1.0 Å. The grid-enclosing box was centered on tacrine and defined to enclose residues located within 5.0 Ǻ. After the simulations were completed, the docked structures were studied and the interactions were analyzed. Hydrogen bond interactions and the binding distance between the donors and acceptors were measured for the best conformer. The AutoDock Vina docking was performed three times. Finally, AutoDock Vina stored the potential energy arising from the interaction of flexible targets with rigid macromolecules. 2D interaction diagrams were generated through BIOVIA Discovery Studio visualizer V17.2. The protein active sites were visualized using the PyMOL visualization tool (The PyMOL Molecular Graphics System, Version 2.0 Schrödinger, LLC.). The hydrophobicity surface views of the protein–ligand complexes were generated using UCSF Chimera X version 1.3.

## Figures and Tables

**Figure 1 plants-12-00059-f001:**
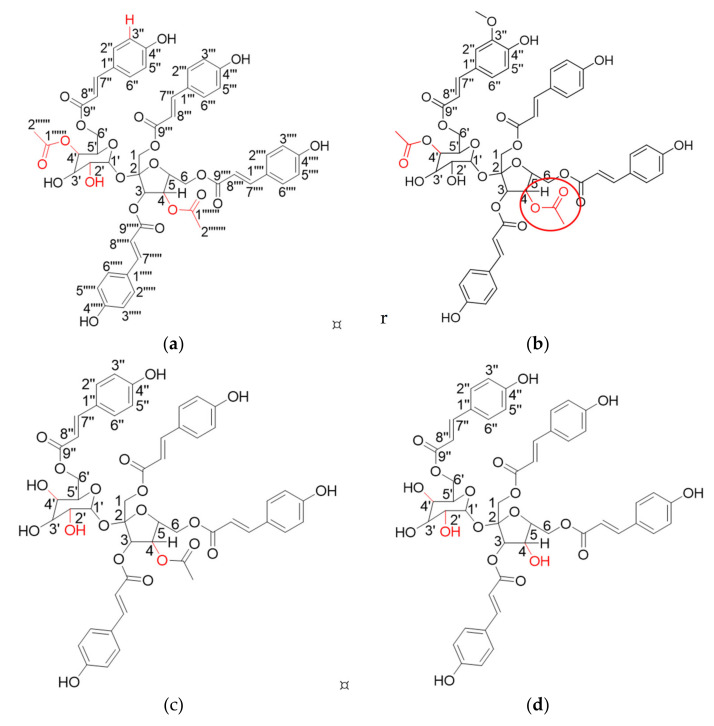
Chemical structure of (**a**) polygonumins A; (**b**) polygonumins B; (**c**) polygonumins C; (**d**) polygonumins D; (**e**) vanicoside A; (**f**) vanicoside B. Circle in blue is an example of a methoxyl group attached at C-3″. Circle in red is an example of an ethanoyl group attached at furanose ring C-4.

**Figure 2 plants-12-00059-f002:**
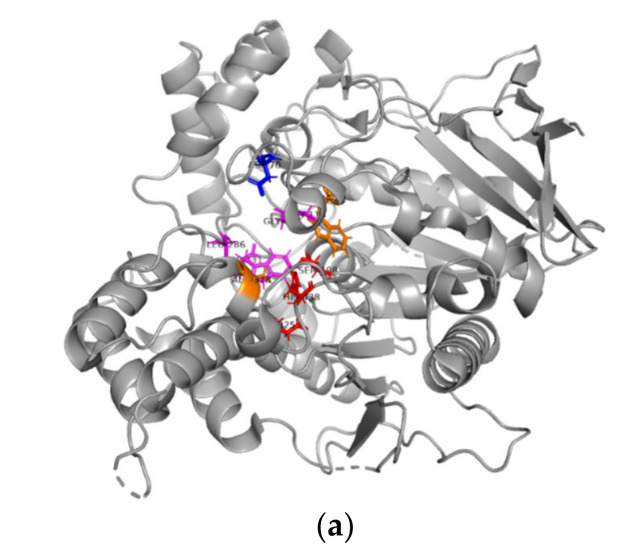
The crystal structure of (**a**) BChE (PDB: 4BDS). The main contributors to the peripheral anionic site (Asp70), the catalytic anionic subsite (Trp82 and Ala328), the catalytic triad (His438, Glu325 and Ser198) and the acyl-binding pocket (Gly116, Gly117, Trp231, Leu286 and Val288) are shown as sticks. The interactive binding pocket obtained from crystal structure of human BChE [46]. (**b**) The crystal structure of AChE (PDB: 2CKM). The main contributors to the peripheral anionic site (Asp72, Tyr121, Trp279, Tyr70, Ser286 and Tyr334: blue), the catalytic anionic subsite (Trp84 and Phe330), the catalytic triad (His440, Glu327 and Ser200) and the acyl-binding pocket (Gly118, Gly119, Phe290 and Phe288) are shown as sticks. The interactive binding site obtained from *Torpedo californica* AChE [47,48]. The crystal structure was obtained from the Protein Data Bank, shown as a gray cartoon with the key amino acids of the active site shown as sticks. The protein structure with active sites was generated and analyzed using the PyMOL visualization tool. The peripheral anionic site is highlighted in blue, the catalytic anionic subsite is highlighted in orange, the catalytic triad is highlighted in red and the acyl-binding pocket is highlighted in magenta.

**Figure 3 plants-12-00059-f003:**
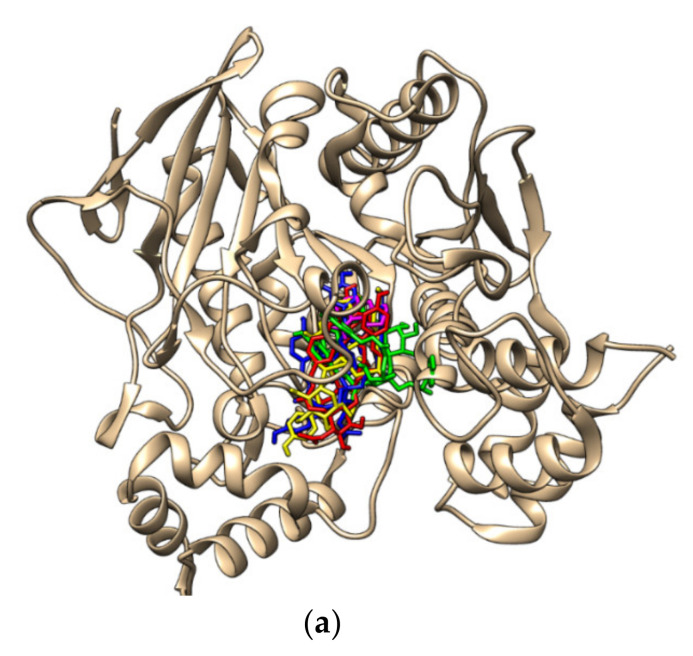
Molecular docking analysis of (**a**) the BChE protein structure: 4BDS and (**b**) the AChE protein structure: 2CKM. (polygonumins A: red stick; polygonumins B: blue stick; polygonumins C: yellow stick; polygonumins D: green stick; tacrine: magenta.)

**Figure 4 plants-12-00059-f004:**
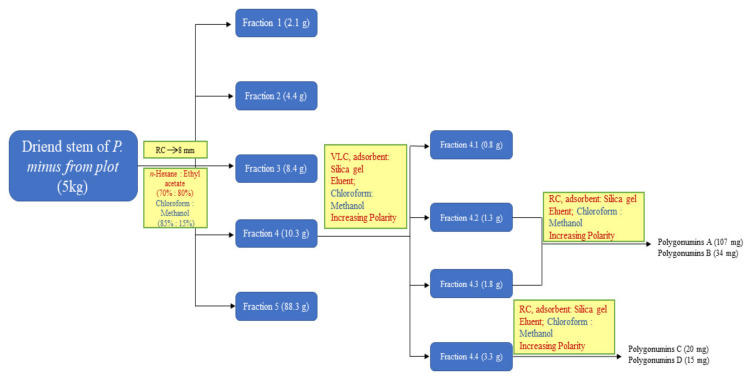
Schematic flow chart of the isolation and extraction procedure of polygonumins A, B, C and D.

**Table 1 plants-12-00059-t001:** Antioxidant activity assayed by DPPH and ABTS.

Substance	DPPH IC_50_ * (µg/mL)	ABTS IC_50_ * (µg/mL)
Polygonumins A	850 ± 6.0	>1000
Polygonumins B	27 ± 1.5	210 ± 5.7
Polygonumins C	240 ± 4.6	460 ± 6.0
Polygonumins D	>1000	>1000
Gallic acid	36 ±1.1	52 ± 2.8

* Data are expressed as mean ± SEM of five independent experiments.

**Table 2 plants-12-00059-t002:** Anti-acetylcholinesterase (AChE) activity and anti-butyrylcholinesterase (BChE) activity of compounds tested at 100 µg/mL.

Substance	Anti-Acetylcholinesterase Activity (AChE) % Inhibition *	Anti-Butyrylcholinesterase Activity (BChE) % Inhibition *
Polygonumins A	15.3 ± 1.5	<10
Polygonumins B	71.8 ± 0.6	36.2 ± 2.1
Polygonumins C	59.2 ± 3.7	42.7 ± 0.6
Polygonumins D	30.3 ± 1.1	37.5 ± 2.3
Tacrine	89.6 ± 0.2	80.9 ± 0.6

* Data are expressed as mean ± SEM of five independent experiments.

**Table 3 plants-12-00059-t003:** Anti-HIV-1 protease activity of compounds tested at 100 µg/mL.

Substance	% Inhibition *
Positive control pepstatin A, 1 mM	88.74 ± 0.4
Polygonumins A	50.45 ± 1.1
Polygonumins B	33.24 ± 2.0
Polygonumins C	91.58 ± 0.2
Polygonumins D	91.38 ± 0.1

* Data are expressed as mean ± SEM of five independent experiments.

**Table 4 plants-12-00059-t004:** Binding energies and active binding sites of the tested compounds in BChE using the AutoDock Vina 4.0 molecular docking program and discovery studio.

Compound Name	Binding Energies kcal mol^−1^	H-Bonds Interaction	Hydrophobic Interaction (π-π)	Interacting Site
Tacrine	−8.2	**His438**,	**Ala328, Trp82**	CT ^a^, CAS ^b^
Polygonumins A	−8.0	**Trp82**, Thr120, **His438**	Tyr332, **Ala328**	CAS, CT
Polygonumins B	−9.4	**Gly117**, Ala199, **Gly116**, Tyr128	**Trp231**, **Trp82**, Val148, Phe329, **Ala328**	ABP ^c^, CAS
Polygonumins C	−9.5	**Gly117**, **Gly116**, Gly439, Glu197, Ser287, Thr120	**Trp82**, **Ala328**, Ala277, Leu286, Phe329	ABP, CAS
Polygonumins D	−9.2	Tyr128, Thr284, Pro 285, Ser198, **Gly116**, Asn68, **Asp70**	Phe329, **Ala328**, **Trp82**, Ala 277, Gly 115	PAS ^d^, ABP, CAS

Highlighted in bold, the compound interacted at the established BChE active gorge [46]. ^a^ Catalytic triad; ^b^ catalytic anionic subsite; ^c^ acyl-binding pocket; ^d^ peripheral anionic site.

**Table 5 plants-12-00059-t005:** Binding energies and active binding sites of the tested compounds in AChE using the AutoDock Vina 4.0 molecular docking program and discovery studio.

Compound Name	Binding Energies kcal mol^−1^	H-Bonds Interaction	Hydrophobic Interaction (π-π)	Interacting Site
Tacrine	−9.3	**Asp72**	**Trp84**, **Phe330**	PAS ^a^, CAS ^b^
Polygonumins A	−7.2	Ile287, Phe 331, Gln 74, Asn 399, Gly 328	**Tyr70**, Pro232	PAS
Polygonumins B	−9.2	Gly118, Gly335, Tyr334, **Ser286** **Trp279**, **Tyr70**, Asp285, Asn280 Asp276, **Trp84**	**Phe330**, **Trp84**, Asn280	PAS, CAS, ABP ^c^
Polygonumins C	−9.4	Ser286, **Trp279**, Gly335, **Tyr334**, Glu73, **Trp84**, Gly118	**Tyr70**, **Tyr121**, **Phe330**	PAS, CAS, ABP
Polygonumins D	−6.9	Trp114, Thr126, **Trp84**, Ser124, Ser81, Glu82, **Tyr334**, Glu73	Leu127	PAS, ABP

Highlighted in bold, the compound interacted at the established AChE active gorge. ^a^ Peripheral anionic site; ^b^ catalytic anionic subsite; ^c^ acyl-binding pocket.

## Data Availability

Data is contained within the article and Appendix A.

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
