# Peer review of "Bioassay Analysis and Molecular Docking Study Revealed the Potential Medicinal Activities of Active Compounds Polygonumins B, C and D from *Polygonum minus* (*Persicaria minor*)"

_plants, 2022, doi:10.3390/plants12010059_

Round 1

Reviewer 1 Report

Dear Authors,

The MS entitled “Bioassays analysis and molecular docking revealed the potential active pure compounds from Polygonum minus: polygonumins B, C and D for Alzheimer diseases and HIV treatments” is original research topic and can be accepted for publishing after minor revision.

Please, check some comments bellow

In Asbtract part instead of present text in L 19-21 would be good to say “ Next pattern of antioxidant activity among polygonumins derivatives was observed in DPPH and ABTS aassays: polygonumins B>polygonumins C>polygonumins A>polygonumins D.

Introduction part

L47 After mention of Polygonum minus would be good to admit that this herb is native to Asia, but also distributed widely in Europe and Australia. Then these research will be also more interesting for more readers.

After L66 It would be good to admit role of molecular docking studies in the discovering of biological active compounds from various herbs with possible anthicholinesterase potential. Its good to say - Nowadays is actively developing molecular docking studies in the discovering of biological active compounds from various herbs with possible anthicholinesterase potential (ref).

Please, see next references bellow

Karakaya S, Bingol Z, Koca M, Dagoglu S, Pınar NM, Demirci B, Gulcin İ, Brestic M, Sytar O. Identification of non-alkaloid natural compounds of Angelica purpurascens (Avé-Lall.) Gilli. (Apiaceae) with cholinesterase and carbonic anhydrase inhibition potential. Saudi Pharm J. 2020 Jan;28(1):1-14. doi: 10.1016/j.jsps.2019.11.001. Epub 2019 Nov 13. PMID: 31920428; PMCID: PMC6950969.

Alqahtani YS. Bioactive stigmastadienone from Isodon rugosus as potential anticholinesterase, α-glucosidase and COX/LOX inhibitor: In-vitro and molecular docking studies. Steroids. 2021 Aug;172:108857. doi: 10.1016/j.steroids.2021.108857.

 A caryophyllene oxide and other potential anticholinesterase and anticancer agent in Salvia verticillata subsp. amasiaca (Freyn & Bornm.) Bornm. (Lamiaceae), Journal of Essential Oil Research, 32:6, 512-525, DOI: 10.1080/10412905.2020.1813212

Author Response

Response to reviewer 1 comments:

The MS entitled “Bioassays analysis and molecular docking revealed the potential active pure compounds from Polygonum minus: polygonumins B, C and D for Alzheimer diseases and HIV treatments” is original research topic and can be accepted for publishing after minor revision.

Response 1: Thank you for your recommendation, comments and suggestion to improve our current manuscript.  We have tried our very best to address all your comments and we hope that the revised manuscript will be accepted for publication.

In Asbtract part instead of present text in L 19-21 would be good to say “ Next pattern of antioxidant activity among polygonumins derivatives was observed in DPPH and ABTS aassays: polygonumins B>polygonumins C>polygonumins A>polygonumins D.

Response 2: We have rephrased this part as suggested by reviewer. Please refer to Page 1 Line 20-24.

Introduction part

L47 After mention of Polygonum minus would be good to admit that this herb is native to Asia, but also distributed widely in Europe and Australia. Then these research will be also more interesting for more readers.

Response 3: We overlooked that this plant is also grown in Europe and part of Australia. As suggested by reviewer, we have added some reference to support that this plant is also found in parts of Europe and Australia. Please refer to Page 2 Line 51-53.

After L66 It would be good to admit role of molecular docking studies in the discovering of biological active compounds from various herbs with possible anthicholinesterase potential. Its good to say - Nowadays is actively developing molecular docking studies in the discovering of biological active compounds from various herbs with possible anthicholinesterase potential (ref).

Response 4: We have rephrased this part as suggested and added the role of molecular docking in drug discovery. Thank you for the references. We have included the references as suggested in our manuscript. Please refer to Page 2 Line 77-81.

Reviewer 2 Report

This MS reports three more phenyl propanoid sucrose esters from the well-studied edible and medicinal herb Polygonum minus/Persicaria minor, and some in vitro biological activities. In my opinion this MS might be suitable for publication, although plants seem the wrong journal, after major revisions to answer the following points:

1.       State that Polygonum minus is a synonym of the accepted name Persicaria minor. State what parts were collected, on what date, and give the herbarium code of the voucher specimen.

2.       This plant is very well-studied with >100 papers. Cite more recent key reviews, e.g. Hamid et al. 2020 and Christapher et al. 2015.

3.       Give original references for distribution and traditional names and uses – not references 4-6.

4.       The trivial name “polygonumins A” has confused the Chemical Abstracts people, who have listed the structure of limonoid “polygonumin A” with the original Ahmad et al. 2018 Chemical Abstract. This should be discussed in the current MS and clarified. Perhaps use vanicoside derived names. Discuss the original report by Zimmermann & Sneden identifying Vanicosides A and B as Protein Kinase C Inhibitors from Polygonum pensylvanicum, show their structures, and review the already published papers on vanicosides/phenyl propanoid sucrose esters.

5.       The assignment of the structure of “polygonumins A” is not acceptably rigorous. How is the 3’’ methoxy phenyl propanoid group placed at 6’ of sucrose rather than at one of the other available positions? A detailed 2D nmr analysis is needed, plus comparisons with vanicoside A.

6.       The legend for Table 1 should state the values given are means of ?n replicates -= +- Standard Deviation (if it is). Round the values to one figure for SD, then the means accordingly.

7.       Give the concentrations tested in Tables 2 and 3. Why no replicates? Delete “high” for these activities from the Abstract, unless tested concentrations were sub micromolar.

8.       Whatever these in vitro activities are, they cannot justify “Alzheimer diseases and HIV treatments” in the title – this would need in vivo results at least in animal models. Delete this phrase.

9.       In my opinion, the molecular modelling/docking section seems far too long for so little in vitro biological data. The original Ahmad et al. 2018 paper covers all this for A – summarise the differences for B C and D and put the rest in supporting info.

Author Response

This MS reports three more phenyl propanoid sucrose esters from the well-studied edible and medicinal herb Polygonum minus/Persicaria minor, and some in vitro biological activities. In my opinion this MS might be suitable for publication, although plants seem the wrong journal, after major revisions to answer the following points:

State that Polygonum minusis a synonym of the accepted name Persicaria minor. State what parts were collected, on what date, and give the herbarium code of the voucher specimen.

Response 1: We have included the synonym name as persicaria minor in the manuscript. We have collected the whole part of the plant from experimental plot and we have given the date, the herbarium code and voucher specimen as requested. However, we extracted the stems parts to produce polygonumins derivatives compound. Please see plant materials section in 4.0- Page 13 Line 390-394.

This plant is very well-studied with >100 papers. Cite more recent key reviews, e.g. Hamid et al. 2020 and Christapher et al. 2015.

Response 2: We have cited a recent publication on polygonum minus. Please refer to 2 Line no 61-64.

Give original references for distribution and traditional names and uses – not references 4-6.

Response 3: We have given the original citation as requested. Please see the corrected citation in the manuscript. Please refer to Page 2 Line 49-56. Reference [4] and [9].

The trivial name “polygonumins A” has confused the Chemical Abstracts people, who have listed the structure of limonoid “polygonumin A” with the original Ahmad et al. 2018 Chemical Abstract. This should be discussed in the current MS and clarified. Perhaps use vanicoside derived names. Discuss the original report by Zimmermann & Sneden identifying Vanicosides A and B as Protein Kinase C Inhibitors from Polygonum pensylvanicum, show their structures, and review the already published papers on vanicosides/phenyl propanoid sucrose esters.

Response 4: We have discussed vanicoside A structure and compared the structure to polygonumins A in Ahmad et al.2018. In this paper, we highlight the structure of polygonumins B,C and D and made comparison with polygonumins A structure. Although the details comparison of vanicoside A and polygonumins A has been mentioned in our previous study, we discussed a brief structural difference between them as suggested by you (Page 2, line 83-91). We also would like to highlight that Polygonumins A has been patent filed and under Malaysia patent protection. Therefore we believed the name should be remained as it is as this is newly compounds isolated from different species of Polygonum spp` as compared to previous reported by Zimmermann et al 1994

The assignment of the structure of “polygonumins A” is not acceptably rigorous. How is the 3’’ methoxy phenyl propanoid group placed at 6’ of sucrose rather than at one of the other available positions? A detailed 2D NMR analysis is needed, plus comparisons with vanicoside A.

Response 5: We have discussed in details the structure of polygonumins A and vanicoside A and has made the comparison in Ahmad et al.2018. We have included the NMR data of polygonumins A in figure S1(supporting information) for your reference.

The NMR data below explained why 3’’ methoxy phenyl propanoid group placed at 6’sucrose. We have made some correction in Table S1

Correction: HSQC for carbon 6' (63.1) bind two signals at 4.37 (m, 1H) and 4.18 (m, 1H)- Please see attachment for further clarification

The legend for Table 1 should state the values given are means of ?n replicates -= +- Standard Deviation (if it is). Round the values to one figure for SD, then the means accordingly.

Response 6: We have stated the SEM values with number of replicates in table 1, table 2 and table 3. We have made correction accordingly.

Give the concentrations tested in Tables 2 and 3. Why no replicates? Delete “high” for these activities from the Abstract, unless tested concentrations were sub micromolar.

Response 7: The concentration of the tested compound was 100µg/ml. When we convert this concentration into µM, the concentration is as follows:

Polygonumins A: 99µM

Polygonumins B: 96µM

Polygonumins C: 103µM

Polygonumins D: 108µM

The concentration of pepstatin A (positive control) in this study was 1mM=1000µM, which was about 10 times higher than our tested compound that means higher than the positive control as stated in the manuscript.

Whatever these in vitro activities are, they cannot justify “Alzheimer diseases and HIV treatments” in the title – this would need in vivo results at least in animal models. Delete this phrase.

Response 8: We followed your suggestion to change the title and removed the sentences ‘’Alzheimer diseases and HIV treatments". We agreed that it must undergo in vivo and clinical trials to be accepted as treatment. However, we did mention the compounds has potential that could be developed further as AD and HIV targeted drugs.

In my opinion, the molecular modelling/docking section seems far too long for so little in vitro biological data. The original Ahmad et al. 2018 paper covers all this for A – summarise the differences for B C and D and put the rest in supporting info.

Response 9: Table 4 and Table 5 has summarized the docking analysis in details (binding energies, H bond interaction, hydrophobic interaction and interacting site) for polygonumins A, B, C and D. Figure 5 (Hydrophobicity surface view of polygonumins derivatives with BChE and AChE protein), figure 6 (two-dimensional illustration showing the interaction of BChE) and figure 7 (two-dimensional illustration showing the interaction of AChE) are excluded in the main manuscript and assigned as supporting information as suggested by you. Please see new supporting information enclosed

Reviewer 3 Report

It is a well-conducted study, with very interesting results and novel molecules, but it needs to be reviewed, so I make the following comments

 Use italics in the scientific names of plants

 Include in the identification number of the species of this study

Supplemental Data. Merge all NMR spectra into one PDF file. Attach the HSQC of the novel compounds as well as the COSY experiment, mass and IR spectra.

 Important note. In the 13C NMR spectrum, a characteristic methoxyl signal is observed (d 55 ppm, Fig S8) and it is not present in the structure that they are proposing, so the chemical structure must be carefully reviewed.

The table is not complete in the supplementary data.

 I suggest putting conclusions

Author Response

It is a well-conducted study, with very interesting results and novel molecules, but it needs to be reviewed, so I make the following comments

Use italics in the scientific names of plants

Response 1: We have corrected scientific names in italic form

 Include in the identification number of the species of this study

Response 2: We have included the identification no. of the species study in materials and method in section 4.1: plant materials. Please refer to Page 13 Line no. 390-394

Supplemental Data. Merge all NMR spectra into one PDF file. Attach the HSQC of the novel compounds as well as the COSY experiment, mass and IR spectra.

Response 3: We have merged all the NMR data in one PDF file for each of the compounds and attached the HSQC and all the information needed.

Important note. In the 13C NMR spectrum, a characteristic methoxyl signal is observed (d 55 ppm, Fig S8) and it is not present in the structure that they are proposing, so the chemical structure must be carefully reviewed.

Response 4: We have included the new NMR data for polygonumins D in supporting information figure S4

The table is not complete in the supplementary data.

Response 5: We have completed the Table S1 as requested

 I suggest putting conclusions

Response 6: We have put the conclusion in the manuscript. Please refer section 3.0 conclusions (Page 8 Line no 299-310)

Round 2

Reviewer 2 Report

This MS reports three more phenyl propanoid sucrose esters from the well-studied edible and medicinal herb Polygonum minus/Persicaria minor, and some in vitro biological activities. I reviewed a previous version of this MS.  In my opinion this MS might be suitable for publication, although plants seem the wrong journal, after revisions to answer the following points:

1.       I repeat that this point should be addressed. The trivial name “polygonumins A” has confused the Chemical Abstracts people, who have listed the structure of limonoid “polygonumin A” with the original Ahmad et al. 2018 Chemical Abstract. This should be discussed in the current MS and clarified. Perhaps use vanicoside derived names. Show the structures of vanicosides A and B, and review the already published papers on vanicosides/phenyl propanoid sucrose esters.

2.       I repeat - Give the concentrations tested in Tables 2 and 3. Delete “high” for these activities from the Abstract, unless tested concentrations were sub micromolar.

Author Response

 I repeat that this point should be addressed. The trivial name “polygonumins A” has confused the Chemical Abstracts people, who have listed the structure of limonoid “polygonumin A” with the original Ahmad et al. 2018 Chemical Abstract. This should be discussed in the current MS and clarified. Perhaps use vanicoside derived names. Show the structures of vanicosides A and B, and review the already published papers on vanicosides/phenyl propanoid sucrose esters.

Response. We have shown the structure of vanicosides A and B on Figure 1 page 4. As suggested, we have comprehensively discussed the similarities and the differences between vanicosides A, B and  polygonumins derivatives based on the published paper. Please see line 83-93 (page 2) and line 145 to 152 (page 5). As the vanicosides were found from different source of Polygonum spp., therefore we believed the polygonumins A and its derivatives name should be remained to show that the compounds were purely isolated from Polygonum minus. We have intensively run NMR and structural analysis to confirm that these compounds have differences as compared to the publication by other researchers. We hope that reviewer will consider this explanation and justification. We really appreciate the comment.

2. I repeat - Give the concentrations tested in Tables 2 and 3. Delete “high” for these activities from the Abstract, unless tested concentrations were sub micromolar.

Response 2: We have stated the concentration tested at the table 2 and table 3 caption. Please refer to page 6. The word 'high' has been deleted in the abstract  as suggested. Please refer to page 1 line 24

Reviewer 3 Report

No comment

Author Response

Dear reviewer,

Thank you for your good comments and suggestion. We appreciate your willingness to review this manuscript.

Round 3

Reviewer 2 Report

This MS reports three more phenyl propanoid sucrose esters from the well-studied edible and medicinal herb Polygonum minus/Persicaria minor, and some in vitro biological activities. I reviewed previous versions of this MS.  In my opinion this MS might be suitable for publication, although plants seem the wrong journal, after revision to answer the following point:

1.       The authors have again ignored my previous suggestion: the trivial name “polygonumins A” has confused the Chemical Abstracts people, who have listed the structure of limonoid “polygonumin A” with the original Ahmad et al. 2018 Chemical Abstract. This should be discussed in the current MS and clarified. Perhaps use vanicoside derived names.

Author Response

Thank you for your great concern on the trivial name. Previously, when we named polygonumins A, we were not aware that a new compound of limonoid group (polygonumin A) was published. When we search using the keyword of  polygonumins A, the search result shows just our manuscript (Ahmed et al 2018). However, when we extend the search by google limonoid-polygonumin A, we were surprised that there is another compound that has been named polygonumin A before. The compound structure is so different from the one that we have isolated. 

Polygonumin A and B belong to limonoid group, whereas our polygonumins are phenylpropanoid sucrose ester, similar to vanicosides.  We have informed our reader in the manuscript that polygonumins A (Ahmad et al. 2018) and polygonumin A ( Liu J, 2001) are from different groups and different polygonum sp.  (page 5 line 153-158). 

Moreover the mass and the NMR spectra of polygonumin is different from our polygonumins.  The difference between these two names is the letter ’s’. The letter ‘s’ means our compound is isolated from polygonum minus. We could not change polygonumins A into another name (vanicoside) as suggested by reviewer due to the argument that we had mentioned in the previous round of response. 

We would like to highlight again that we have published our compound as polygonumins A isolated from polygonum minus in 2018. Moreover, the compound has been patented in Malaysia. We hope that you will consider our answer and thank you for highlighting this issue to us.
